Overexpression of fucosyltransferase 8 reverses the inhibitory effect of high-dose dexamethasone on osteogenic response of MC3T3-E1 preosteoblasts

Wu Zhiming 1 2 3 4
Lin Tianye 1 2 3
Kang Pan 1 2 3
Zhuang Zhikun 5
Wang Haibin 1 2 3
He Wei 1 6
Wei Qiushi 1 6
Li Ziqi ionsev0214@126.com 1 6
1 Guangzhou University of Chinese Medicine , Guangzhou , China
2 Department of Joint Orthopaedic, The First Affiliated Hospital of Guangzhou University of Chinese Medicine , Guangzhou , China
3 Lingnan Medical Research Center of Guangzhou University of Chinese Medicine , Guangzhou , China
4 Shenzhen Hospital (Futian) of Guangzhou University of Chinese Medicine , Shenzhen City , Guangdong Province , China
5 Department of Joint Orthopaedic, Quanzhou Orthopedic-Traumatological Hospital of Fujian Traditional Chinese Medicine University , Quanzhou , China
6 Department of Joint Orthopaedic, The Third Affiliated Hospital, Guangzhou University of Chinese Medicine , Guangzhou , China
Etemad-Moghadam Shahroo
Electronic publication date: 2021 Dec 9
Publication date: 2021
Volume: 9
Electronic Location ID: e12380
Received 2021 Jul 22; Accepted 2021 Oct 4
Copyright: ©2021 Wu et al.
Copyright year: 2021
Copyright holder: Wu et al.
License: This is an open access article distributed under the terms of the Creative Commons Attribution License, which permits unrestricted use, distribution, reproduction and adaptation in any medium and for any purpose provided that it is properly attributed. For attribution, the original author(s), title, publication source (PeerJ) and either DOI or URL of the article must be cited.
License URL: https://creativecommons.org/licenses/by/4.0/

Keywords: Dexamethasone, Osteogenesis, Fucosyltransferase 8, Transforming growth factor beta

Funding: National Natural Science Foundation of China 81904226 Traditional Chinese Medicine Bureau of Guangdong Province 20201098 This work was supported by the National Natural Science Foundation of China (grant number 81904226 to Ziqi Li) and the Traditional Chinese Medicine Bureau of Guangdong Province (grant number 20201098 to Ziqi Li). The funders had no role in study design, data collection and analysis, decision to publish, or preparation of the manuscript.

==============================
Background

Core fucosylation catalyzed by FUT8 is essential for TGF-β binding to TGF-β receptors.

Methods

Indirect TGF-β1 binding assay was used to evaluate the ability of TGF-β1 to bind to TGFBRs, Alizarin red and alkaline phosphatase staining were used to detect osteogenic differentiation and mineralization ability , western blot and quantitative RT-PCR were used to measure the differential expression of osteogenesis-related proteins and genes. Plasmid-mediated gain-of-function study. The scale of core fucosylation modification was detected by Lectin-blot and LCA laser confocal.

Results

Our results showed that compared with vehicle treatment, high-dose (10−6 and 10−5 M) dexamethasone significantly inhibited cell proliferation, osteogenic differentiation, and FUT8 mRNA expression while promoting mRNA expression of adipogenesis-related genes in MC3T3-E1 cells, suggesting that downregulation of FUT8 is involved in the inhibitory effect of high-dose dexamethasone on osteogenesis. Overexpression of FUT8 significantly promoted osteogenic differentiation and activated TGF-β/Smad signaling in MC3T3-E1 cells in the presence of high-dose dexamethasone, suggesting that FUT8 reverses the inhibitory effect of high-dose dexamethasone on osteogenesis. In addition, lectin fluorescent staining and blotting showed that overexpression of FUT8 significantly reversed the inhibitory effects of high-dose dexamethasone on core fucosylation of TGFBR1 and TGFBR2. Furthermore, indirect TGF-β1 binding assay showed that overexpression of FUT8 remarkably promoted TGF-β1 binding to TGFBRs in MC3T3-E1 cells in the presence of high-dose dexamethasone.

Conclusions

Taken together, these results suggest that overexpression of FUT8 facilitates counteracting the inhibitory effect of dexamethasone on TGF-β signaling and osteogenesis.

Introduction

Glucocorticoids, such as hydrocortisone and dexamethasone, are potent anti-inflammatory agents that are widely used in the treatment of various disorders, such as autoimmune diseases, bronchial asthma, and transplant rejection.  However, high-dose and long-term glucocorticoid treatment may lead to osteoporosis and osteonecrosis (Kerachian, Seguin & Harvey, 2009) by promoting apoptosis of osteoblasts and mature osteocytes, extending the life span of osteoclasts, as well as suppressing proliferation and terminal differentiation of preosteoblasts (Carcamo-Orive et al., 2010). Glucocorticoid-induced osteoporosis occur in about 30–50% long-term users of glucocorticoids, representing the most common cause of secondary osteoporosis (Hayat & Magrey, 2020; Staa, Leufkens & Cooper, 2002). Studies have shown that the dose and duration of glucocorticoid treatment determine bone fragility (Barturen et al., 2020) and that high-dose and long-term intake of glucocorticoids is a major contributor to osteoblast and osteocyte apoptosis (Hsu & Nanes, 2017; Weinstein, 2011). Thus, the identification of key molecules involved in glucocorticoid-induced osteoporosis is urgently needed for improving bone regeneration in glucocorticoid therapies.

The transforming growth factor beta (TGF-β)/TGF-β receptors (TGFBRs)/Smads signaling pathway activates osteogenesis through type I and type II serine/threonine kinase receptors TGFBR1 and TGFBR2 (Nguyen et al., 2013; Wu, Chen & Li, 2016). TGFBR1 is phosphorylated after the activation of TGFBR2 (Wagner et al., 2010), which initiates intracellular signaling through phosphorylation of receptor-regulated Smads (R-Smads). The activated R-Smads form complexes with common-partner Smads (co-Smads) and Smad4 and then translocate into the nucleus to direct transcriptional response (Yi et al., 2010). Studies have shown that TGF-β/TGFBRs/Smads signaling is inhibited in osteoporosis and osteonecrosis of the femoral head induced by glucocorticoids (Li et al., 2017; Tao et al., 2017; Xie, Hu & Shi, 2018); however, the underlying mechanism remains unclear.

Core fucosylation catalyzed by α-1,6 fucosyltransferase 8 (FUT8) is the transfer of fucose from GDP-fucose to the core structure of the N-glycan chain (Calderon, Zhou & Guan, 2017). Abnormal core fucosylation contributes to multiple pathological conditions, such as cancer, pulmonary emphysema, and tissue remodeling (Bastian et al., 2021; Iijima et al., 2017; Wang et al., 2017). However, the association of core fucosylation with bone formation remains unclear. It has been reported that FUT8-mediated core fucosylation is crucial for TGF-β binding to TGFBRs. FUT8 deficiency-induced deficit of core fucosylation results in impaired binding of TGF-β to TGFBRs, decreased TGFBR activation, as well as inhibited downstream signaling (Schachter, 2005; Venkatachalam & Weinberg, 2013). Our previous study has shown that fucosylation is downregulated in glucocorticoid-induced osteonecrosis of the femoral head (Song et al., 2018), suggesting that FUT8-mediated core fucosylation may play a role in osteogenesis. Therefore, we hypothesized that glucocorticoids, such as dexamethasone, might inhibit osteogenic differentiation of preosteoblasts by impairing the core fucosylation and that overexpression of FUT8 might reverse the inhibitory effect of dexamethasone on osteogenic differentiation by activating TGF-β/TGFBRs/Smads signaling.

Mouse calvaria-derived osteoblast precursor cell line MC3T3-E1 possesses the majority of the molecular features of osteocytes (Bhalerao et al., 1995) and has been widely used to study the effect of TGF-β signaling on osteogenesis (Chen et al., 2019; Li & Jiang, 2019; Wang et al., 2012). In this study, we investigated the role and the underlying mechanism of FUT8 in the regulation of osteogenic response in MC3T3-E1 cells exposed to high-dose dexamethasone. Our findings suggest that overexpression of FUT8 may counter the inhibitory effect of high-dose dexamethasone on osteogenic response of preosteoblasts, serving as a potential therapeutic strategy for promoting bone formation in glucocorticoid therapy.

Materials & Methods

Cell culture and dexamethasone treatment

Murine osteoblast precursor cell line MC3T3-E1 was obtained from Shanghai Zhong Qiao Xin Zhou Biotechnology (Shanghai, China). Cells were maintained in α-MEM (Gibco, Gaithersburg, MD, USA) supplemented with 10% fetal bovine serum (FBS), 50 µM ascorbate, 1% penicillin-streptomycin, and 10 mM β-glycerophosphate in a humidified atmosphere with 5% CO2 at 37 °C. Cells at the third or fourth passage were used in the experiments. For dexamethasone treatment, cells were cultured in α-MEM supplemented with 1% FBS for 24 h and then stimulated with different concentrations of dexamethasone (0, 10−8, 10−7, 10−6, and 10−5 M; Macklin, Shanghai, China) (Belka, Nickel & Kurth, 2019).

Plasmid construction and transfection

Mouse FUT8 was amplified using primers 5′-TACAAGTCCGGACTCAGATCTGCCACCA TGCGGGCATGGACTGGT-3′ (sense) and 5′-GTACCGTCGACTGCAGAATTCCTATTT TTCAGCTTCAGGATATGTGG-3′ (antisense) and cloned into pEGFP-C1 vector (Clontech, Mountain View, CA, USA). Mouse TGF-β1 was amplified using primers 5′-CACTAGTCCAGTGTGGGAATTCGCCACCATGCCGCCCTCGGGGCTG (sense)-3′ and 5′-GCCCTCTAGACTCGAGCGGCCGCTCAGCTGCACTTGCAGGAGC-3′  (antisense) and cloned into pcDNA3.1-V5-HisB vector (V81020; Invitrogen, Carlsbad, CA, USA). The clones were confirmed by DNA sequencing. MC3T3-E1 cells were transiently transfected with corresponding vectors using Lipofectamine® 3000 (Invitrogen).

Cell proliferation assay

Cell proliferation was examined using the cell counting kit-8 (CCK8) assay. Briefly, MC3T3-E1 cells were seeded in a 96-well plate at a density of 1,000 cells/well and cultured for 24 h at 37 °C. Cells were treated with different concentrations of dexamethasone as above mentioned. Phosphate buffered saline (PBS) was used as a vehicle. At 24 h, 48 h, 72 h, 7 d, or 14 d after treatment, cells were incubated with CCK8 solution (10 µL/well) for 1 h at 37 °C. The absorbance was measured at 450 nm wavelength using a Labserv K3 microplate reader (Thermo Fisher Scientific, Waltham, MA, USA).

Alkaline phosphatase staining

MC3T3-E1 cells were cultured in osteogenic medium containing 50 µM ascorbate and 10 mM β-glycerophosphate for 7 or 14 days, washed with ice-cold PBS, and fixed with 95% ice-cold ethanol for 30 min. Alkaline phosphatase (ALP) staining was performed using an ALP staining kit (Beyotime, Guangzhou, China) following the manufacturer’s instruction. ALP staining was observed using a 450-fluorescent inverted phase-contrast microscope (Olympus, Tokyo, Japan)

Alizarin red staining

Alizarin red staining was performed to examine mineralization of MC3T3-E1 cells. Cells were cultured in osteogenic medium for 7 or 14 days and then fixed in 4% paraformaldehyde (Sigma-Aldrich, St. Louise, MO, USA) for 30 min. After 3 washes with ice-cold PBS, cells were stained with Alizarin red (Sigma-Aldrich) for 5 min. Alizarin red staining was observed using an Olympus 450-fluorescent inverted phase-contrast microscope.

Quantitative real-time PCR (qRT-PCR)

Total RNA was isolated from MC3T3-E1 cells using Trizol reagent (Invitrogen). qRT-PCR was performed using a SYBR Green PCR kit (Invitrogen) and gene-specific primers (Table 1). ACTB was used as an internal reference. The relative gene expression was determined using the 2−△△Ct method. The PCR reactions were performed in triplicates.

Table 1 Primers for quantitative real-time PCR.

Gene	Forward (5′–3′)	Reverse (5′–3′)	
ACTB	GAGGTATCCTGACCCTGAAGTA	CACACGCAGCTCATTGTAGA	
FUT8	GTGGATGGGAGACTGTGTTTAG	GAGCTCGACCACTTGAATGT	
ALP	CTTTCGTAGCAGCAGCAAAC	GGAGCGCGTCTTGGATATT	
OCN	GGAGCTGCTTTGGTGAGATTAG	GAGTAGCCCAGACTACGGATATT	
OSX	TGGAGAGGGAAAGGGATTCT	GAAATCTACGAGCAAGGTCTCC	
RUNX2	TGGCTTGGGTTTCAGGTTAG	GGTTTCTTAGGGTCTTGGAGTG	
BMP2	ACACAGCTGGTCACAGATAAG	CTTCCGCTGTTTGTGTTTGG	
HDAC5	AGTACCACACCCTGCTCTAT	AGCATGGCGTACATCTTCTG	
PPARγ	CTGGCCTCCCTGATGAATAAAG	AGGCTCCATAAAGTCACCAAAG	
CEBPα	CTCCCAGAGGACCAATGAAATG	TTAGCCGGAGGAAGCTAAGA	
TGFβ1	GGTGGTATACTGAGACACCTTG	CCCAAGGAAAGGTAGGTGATAG	
TGFβ2	GGCTTTCATTTGGCTTGAGATG	CTTCGGGTGAGACCACAAATAG	
TGFβ3	CGCTACATAGGTGGCAAGAA	CAAGTTGGACTCTCTCCTCAAC	
Smad2	GCTGAGTGCCTAAGTGATAGTG	TACAGCCTGGTGGGATCTTA	
Smad3	CGCTGTTCCAGTGTGTCTTA	GATGGAGTTCTCTTCCAAGGTC	
Smad4	CTGTCTTCATCCGGTCTTCATC	GTTGTTCCTGCTCCACTCAT	
TβRI	CCTTGAGTCACTGGGTGTTATG	CCACTTAGCTGTCACCCTAATC	
TβRII	GTTCGTGAGCATGGAGAGATAG	CAGGGCTGAGATGATAAGAGTG	

Western blot analysis

Proteins were isolated from MC3T3-E1 cells using RIPA buffer. A total of 60 µg protein samples were separated using10% SDS-PAGE and transferred to a polyvinylidene difluoride (PVDF) membrane. The membrane was incubated with primary antibody against Smad2/Smad3 (1:1,000; 5678S, Cell Signaling Technology), p-Smad2(Ser465/467)/Smad3 (Ser423/425) (1:1,000; 8828S, Cell Signaling Technology), and β-actin (1:1,000; 4970T, Cell Signaling Technology) Incubated overnight at 4 °C. The membrane was washed three times with TBST and incubated with HRP-linked secondary antibody (1:1,000; 7074P2; Cell Signaling Technology) for 1 h at room temperature.

Lectin blot analysis

Lens Culinaris Agglutinin (LCA) lectin blot analysis was performed to detect the α1,6 fucosylated trimannose-core structure of N-linked oligosaccharide (Imai-Nishiya et al., 2007). Proteins were isolated from cells using RIPA buffer. A total of 60 µg proteins per sample were separated by 10% SDS-PAGE and transferred to a PVDF membrane. The membrane was blocked using 5% fat-free milk in TBST at room temperature overnight, followed by 3 washes with TBST. Then, the membrane was incubated with 3 µg/mL biotinylated-LCA (1:1000; B-1045; Maravai Life Sciences, San Diego, CA, USA) in 5% fat-free milk overnight at 4 °C. After 3 washes with TBST, the membrane was incubated with HRP-conjugated streptavidin (1:1,000; E030100-01; EathOx life science, Millbrae, CA, USA) for 1 h at room temperature. After an incubation with Western blot substrate solution for 2 min, the images were acquired in a darkroom. Data was analyzed using ImageJ software (National Institutes of Health, Bethesda, MD, USA).

Co-immunoprecipitation (Co-IP)

Co-IP was performed using a Beaver Beads protein A/G immunoprecipitation kit (22202-20; Beaver Nano-Technologies, China) following the manufacturer’s instruction. Briefly, cell lysates were centrifuged at 12,000 rpm for 20 min at 4 °C. The supernatant was collected, and 500 µg proteins were incubated with 2 µg anti-TGFBR1 (ab235178; Abcam, Cambridge, UK) or anti-TGFBR2 antibody (ab269279; Abcam)-conjugated protein A/G agarose beads overnight at 4 °C. The immunoprecipitates were collected and washed three times with lysis buffer. Equal amounts (10 µg/lane) of proteins were subjected to 12% SDS-PAGE for LCA blot analysis.

Immunofluorescence staining

MC3T3-E1 cells were seeded in a 6-well plate at a density of 2  ×105 cells/well. After treatment, cells were washed three times with PBS, followed by fixation with 4% paraformaldehyde for 15 min. Cells were then treated with 0.1% PBS-Triton X-100 for 5 min and incubated with rhodamine-conjugated LCA (1:1,000; RL-1042-5; Vector Laboratories, Burlingame, CA, USA) for 1 h at 26–28°C. The nucleus was stained with DAPI (S2110; Solarbio) for 1 min. The fluorescent staining was observed using an Olympus CKX53 inverted microscope. Images were acquired using an UPlanFLN objective at magnification 40 × and analyzed using VistarImage 3.0 software (Nikon, Japan).

Indirect TGF-β1 binding assay

The pcDNA3.1-V5-HisB vector overexpressing His-TGF-β1 was constructed as above mentioned and transfected into MC3T3-E1 cells. The conditioned medium containing His-TGF-β1 was collected at 48 h after transfection. MC3T3-E1 cells transfected with FUT8-overexpressing vectors or empty vectors were incubated with the conditioned medium for 4 h at 4 °C, followed by cell lysis using the lysis buffer containing 25 mM Hepes, 150 mM NaCl, 5 mM EDTA, 10 µg/ml aprotinin, 5 µg/ml leupeptin, 10% glycerol, and 1% Triton X-100 (pH 7.6). Cell lysates were collected by centrifugation at 10,000× g for 15 min. The binding of His-TGF-β1 and endogenous TGFBRs in MC3T3-E1 cells was detected using an ALP-conjugated anti-His antibody (ab49746, Abcam). The bound ALP was measured using a p-nitrophenyl phosphate substrate (Sigma-Aldrich) to quantify the binding of His-TGF-β1 and TGFBRs (Tu et al., 2017).

Statistical analysis

Data are expressed as the mean ± standard deviation. Statistical analysis was performed using SPSS 23.0 software (SPSS Inc., Chicago, IL, USA). Comparisons among different groups were performed using one-way analysis of variance followed by Tukey’s post-hoc test. A P value less than 0.05 was considered statistically significant.

Results

High-dose dexamethasone inhibits cell proliferation, osteogenic differentiation, and FUT8 mRNA expression of MC3T3-E1 cells

To investigate the effect of dexamethasone on preosteoblast proliferation, we treated MC3T3-E1 cells with different concentrations of dexamethasone and examined the cell viability at different time points. The results of CCK8 assay showed that compared with vehicle treatment, low-dose dexamethasone (10−8 and 10−7 M) significantly promoted cell proliferation in a time-dependent manner, whereas high-dose dexamethasone (10−6 and 10−5 M) significantly inhibited cell proliferation with maximum effects at 7 days after treatment in MC3T3-E1 cells (Fig. 1A). Then, we examined the effects of dexamethasone on osteoblast differentiation and mineralization of MC3T3-E1 cells. The results of ALP staining (Fig. 1B, upper panel) and Alizarin red staining (Fig. 1B, lower panel) showed that compared with control cells, cells treated with low-dose dexamethasone exhibited stronger ALP and Alizarin red staining, whereas those treated with high-dose dexamethasone demonstrated weaker staining (Fig. 1B). These data suggest that low-dose dexamethasone promotes, whereas high-dose dexamethasone inhibits cell proliferation, osteoblast differentiation, and mineralization of preosteoblasts.

Figure 1 The effects of dexamethasone on cell proliferation and osteogenic response of MC3T3-E1 cells.

(A) MC3T3-E1 cells were cultured in osteogenic medium containing different doses of dexamethasone (0, 10−8, 10−7, 10−6, and 10−5 M). Cell counting kit-8 assay was performed to measure cell viability at 1 d, 2 d, 3 d, 7 d, and 14 d after treatment. (B) MC3T3-E1 cells were cultured in osteogenic medium containing different doses of dexamethasone as indicated. Alkaline phosphatase (ALP) staining and Alizarin red staining were conducted to examine osteogenesis and mineralization, respectively. C1: control cells cultured in osteogenic medium without dexamethasone; C2: Control cells cultured in medium without osteogenic supplements. (C) MC3T3-E1 cells were cultured in osteogenic medium containing different doses of dexamethasone as indicated. Quantitative real-time PCR (qRT-PCR) was performed at 3 d after treatment to measure mRNA expression of FUT8 as well as osteogenesis- and adipogenesis-related genes. ACTB was used as an internal reference. Data are expressed as the mean ± standard deviation (SD). *P < 0.05, **P < 0.01, ***P < 0.001 vs. 0 M. All experiments were independently repeated at least three times. Dex, Dexamethasone; FUT8, fucosyltransferase 8; ALP: alkaline phosphatase.

To further examine the effect of dexamethasone treatment on osteogenesis and the involvement of core fucosylation, we determined the mRNA levels of FUT8 as well as osteogenesis- and adipogenesis-related genes in MC3T3-E1 cells exposed to different doses of dexamethasone. To avoid the substantial changes in cell numbers resulting from long-term exposure to dexamethasone (Fig. 1A), we treated the cells for 3 day. The results showed that low-dose dexamethasone significantly upregulated mRNA expression of FUT8 whereas high-dose dexamethasone dramatically downregulated mRNA expression of FUT8 (Fig. 1C, upper left panel). Low-dose dexamethasone also significantly upregulated mRNA expression of osteogenesis-related genes, including ALP, BMP2, Osx, Ocn, and Runx2, whereas high-dose dexamethasone generally showed no effects on mRNA expression of these gens (Fig. 1C, upper and middle panels). In contrast, low-dose dexamethasone generally downregulated mRNA expression of adipogenesis-related genes, including CEBPα, HDAC5, and PPARγ, whereas high-dose dexamethasone generally, significantly upregulated these gens (Fig. 1C, lower panel). These data suggest that high-dose dexamethasone supports adipogenesis but inhibits osteogenesis, involving the downregulation of FUT8 expression.

Overexpression of FUT8 promotes osteogenic differentiation of MC3T3-E1 cells regardless of the presence or absence of high-dose dexamethasone

To examine whether compensatory expression of FUT8 could reverse the effect of high-dose dexamethasone on osteogenesis, we overexpressed FUT8 in MC3T3-E1 cells and then treated the cells with vehicle or high-dose dexamethasone. As shown in Fig. 2A, compared with empty vector transfection, overexpression of FUT8 enhanced the intensities of ALP and Alizarin red staining, in the absence of dexamethasone. High-dose dexamethasone treatment resulted in reductions in ALP and Alizarin red staining in empty vector-transfected cells. Importantly, overexpression of FUT8 enhanced ALP and Alizarin red staining in the presence of high-dose dexamethasone, suggesting that overexpression of FUT8 abrogates the inhibitory effects of high-dose dexamethasone on osteogenic differentiation and mineralization of preosteoblasts. Then, we measured the mRNA levels of osteogenic markers in MC3T3-E1 cells in response to FUT8 overexpression. As shown in Fig. 2B, compared with control, FUT8 overexpression dramatically promoted mRNA expression of osteogenic markers, regardless of the presence or absence of dexamethasone. Collectively, the combined results of Figs. 1 and 3 suggest that overexpression of FUT8 reverses the effects of high-dose dexamethasone on osteogenesis.

Figure 2 Overexpression of FUT8 promoted osteogenic differentiation of MC3T3-E1 cells regardless of the presence or absence of high-dose dexamethasone.

MC3T3-E1 cells were transiently transfected with empty or FUT8-overexpressing vectors and incubated in osteogenic medium in the absence or presence of high-dose dexamethasone for 5 or 14 days (A) ALP staining and Alizarin red staining were performed to examine osteogenesis and mineralization, respectively (B) qRT-PCR was performed at 3 days after treatment to measure mRNA expression of osteogenesis-related genes. ACTB was used as an internal reference. Data are expressed as the mean ± SD. *P < 0.05, **P < 0.01, ***P < 0.001, ****P < 0.0001 vs. pEGFP-C1. All experiments were independently repeated at least three times. FUT8, fucosyltransferase; ALP, alkaline phosphatase; Dex, dexamethasone.

Figure 3 The effects of overexpression of FUT8 on TGF/Smad signaling in MC3T3-E1 cells.

MC3T3-E1 cells were transfected with empty or FUT8-overexpressing vectors and cultured in osteogenic medium containing different doses of dexamethasone (0, 10−6, and 10−5 M) for 3 days. (A–C) qRT-PCR was performed to measure mRNA expression of the components of the TGFβ/Smad pathway. ACTB was used as an internal reference. Data are expressed as the mean ± SD. *P < 0.05, **P < 0.01, ***P < 0.001, ****P < 0.0001 vs. pEGFP-C1. (D) Western blot analysis was conducted to measure protein expression of phosphorylated and total Smad2/3 protein levels. β-actin was used as an internal reference. All experiments were independently repeated at least three times. FUT8, fucosyltransferase; Dex, dexamethasone.

Overexpression of FUT8 activates TGF-β/Smad signaling in MC3T3-E1 cells in the presence of high-dose dexamethasone

Considering the critical role of TGF-β/Smad signaling in osteogenesis and the essential role of core fucosylation in TGF-β signaling activation, we examined the effect of FUT8 overexpression on TGF-β/Smad signaling in MC3T3-E1 preosteoblasts exposed to high-dose dexamethasone. qRT-PCR revealed that in general, overexpression of FUT8 remarkably enhanced mRNA expression of TGF-β1, TGF-β2, Smad2, Smad3, Smad4, and TGFBR2, while attenuating mRNA expression of TGF-β3 and TGFBR1 (Figs. 3A–3C), regardless of the presence and absence of dexamethasone. Of note, overexpression of FUT8 induced opposite effects between mRNA expression of TGF- β1/2 and TGF-β3 (Fig. 3A) as well as between mRNA expression of TGFBR1 and TGFR2 (Fig. 3C). In addition, overexpression of FUT8 did not affect the expression of TGF-β2 or Smad4 in the presence of 10−6 M dexamethasone or the expression of TGF-β3 or TGFBR1 in the presence of 10−5 M dexamethasone (Figs. 4A and 4C), suggesting that the concentration of dexamethasone may play a role in FUT8-mediated regulation of TGF β/Smad signaling.

Figure 4 Overexpression of FUT8 promoted core fucosylation of TGFBRs and TGF-β1 binding to TGFBRs in MC3T3-E1 cells in the presence of high-dose dexamethasone.

MC3T3-E1 cells were transfected with empty or FUT8-overexpressing vectors and cultured in osteogenic medium containing different doses of dexamethasone (0, 10−6, and 10−5 M) as indicated for 3 days. (A) Lens culinaris agglutinin (LCA)-based immunofluorescence staining was performed to detect α1,6-fucosylated trimannose-core structure in MC3T3-E1 cells. Scale bar = 100 µm. (B) LCA blot analysis was performed to determine the level of core fucosylated-proteins. (C) Co-immunoprecipitation was conducted to measure the levels of core fucosylated-TGFBR1 and TGFBR2. (D) MC3T3-E1 cells (from A) were transfected with empty or His-tagged TGF-β1 (His-TGF-β1)-overexpressing vectors. The conditioned medium was collected at 2 days after transfection. MC3T3-E1 cells were cultured in the conditioned medium for 4 h, and the cell lysates were collected. The His-TGF-β1 protein bound to endogenous TGFBRs were detected by the ALP-conjugated anti-His antibody and quantified by measuring the bound ALP using a p-nitrophenyl phosphate substrate. Data are expressed as the mean ± SD. *P < 0.05, **P < 0.01, ***P < 0.001, ****P < 0.0001 vs. empty vector. mOD405, milli-absorbance units at 405 nm. All experiments were independently repeated at least three times.

To further investigate whether FUT8 activates TGF-β/Smad signaling, we determined the phosphorylation status of Smad2/3 in MC3T3-E1 cells overexpressing FUT8. As shown in Fig. 3D, compared with empty vector transfection, overexpression of FUT8 markedly elevated the protein levels of total Smad2/3 but reduced phosphorylated Smad2/3 (p-Smad2/3) in the absence of dexamethasone. On the other hand, treatment with high-dose dexamethasone significantly reduced protein levels of total and phosphorylated Smad2/3, suggesting that high-dose dexamethasone suppresses Smad signaling in preosteroblasts. Importantly, in the presence of high-dose dexamethasone, overexpression of FUT8 substantially elevated the protein levels of total and p-Smad2/3, compared with empty vector transfection (Fig. 3D). These results suggest that overexpression of FUT8 reverses the inhibitory effect of high-dose dexamethasone on TGF-β/Smad2/3 signaling in preostroblasts.

Overexpression of FUT8 promotes core fucosylation of TGFBRs and TGF-β1 binding to TGFBRs in MC3T3-E1 cells in the presence of high-dose dexamethasone

Next, we detected α1,6-fucosylated trimannose-core structure in MC3T3-E1 cells to investigated whether FUT8-mediated core fucosylation affects TGF-β1/TGFBRs signaling of preosteoblasts. LCA immunofluorescent staining showed that overexpression of FUT8 substantially promoted core fucosylation compared with the control, whereas high-dose dexamethasone treatment significantly suppressed core fucosylation of MC3T3-E1 preosteoblasts. Of note, overexpression of FUT8 partially but significantly counteracted the suppressing effect of high-dose dexamethasone on core fucosylation (Fig. 4A). Consistent results were observed in LCA blot analysis (Fig. 4B). These data suggest that overexpression of FUT8 reverses the suppressing effect of high-dose dexamethasone on core fucosylation in preosteoblasts.

To identify the downstream molecule of FUT8 in preosteoblasts exposed to high-dose dexamethasone, we detected fucosylated-TGFBR1 and TGFBR2 in MC3T3-E1 cells. The results of Co-IP revealed that compared with empty vector transfection, overexpression of FUT8 dramatically elevated the TGFBR1-LCA/TGFBR1 ratio but not TGFBR2-LCA/TGFBR2 ratio in MC3T3-E1 cells. Conversely, 10−5 M and 10−6 M dexamethasone significantly reduced TGFBR1-LCA/TGFBR1 ratio and TGFBR2-LCA/TGFBR2 ratio, respectively, in MC3T3-E1 cells, compared with vehicle treatment. Importantly, overexpression of FUT8 significantly reversed the effects of high-dose dexamethasone on the ratios of TGFBR1-LCA/TGFBR1 and TGFBR2-LCA/TGFBR2 (Fig. 4C). This finding suggests that overexpression of FUT8 counteracts the inhibitory effect of high-dose dexamethasone on core fucosylation of TGFBR1 and TGFBR2, thereby promoting osteogenesis by activating TGF-β signaling in preosteoblasts.

To further confirm the enhancive role of FUT8 in TGF-β signaling, we examined the change in TGF-β1 binding to TGFBRs in response to FUT8 overexpression. As shown in Fig. 4D, His-TGF-β1 overexpression resulted in a remarkable increase in ALP signal, suggesting that the intensity of ALP signal may reflect the level of His-TGF-β1 binding to TGFBRs. Therefore, we showed His-TGF-β1 overexpression normalized with the empty vector (His-TGF-β1 relative to vector). In the absence of dexamethasone, compared with empty pEGFP-C1 vector transfection, overexpression of FUT8 dramatically elevated the intensity of ALP signal, suggesting that overexpression of FUT8 promotes His-TGF- β1 binding to TGFBRs. In contrast, in empty pEGFP-C1 vector-transfected cells, high-dose dexamethasone significantly reduced the intensity of ALP signal compared with vehicle treatment, suggesting that high-dose dexamethasone inhibits His-TGF-β1 binding to TGFBRs. Of note, overexpression of FUT8 dramatically reversed the inhibitory effects of high-dose dexamethasone on His-TGF-β1 binding to TGFBRs. Together, these data suggest that FUT8 may promote TGF-β1 binding to TGFBRs and then activate downstream signals, facilitating osteogenesis in the presence of high-dose dexamethasone.

Discussion

Glucocorticoids exert dual effects on skeletal metabolism and osteogenesis (Hartmann et al., 2016). Glucocorticoids at physiological levels promote osteoblast proliferation and osteogenic differentiation. In contrast, glucocorticoids at pharmacological doses may induce apoptosis of osteoblasts and osteocytes while inhibiting proliferation and differentiation of osteoprogenitor cells (Phillips et al., 2006). Thus, different doses of dexamethasone may exert different effects on osteogenesis. A recent study has found that treatment with dexamethasone (100 nM) for 6 h significantly downregulates FUT8 expression in human corneal epithelial cells (Kadmiel et al., 2016). In addition, our previous study has demonstrated that fucosylation is downregulated in the glucocorticoid-induced osteonecrosis of the femoral head (Song et al., 2018). These findings suggest that glucocorticoid treatment is associated with dysregulated fucosylation, possibly playing an important role in glucocorticoid-induced imbalance in bone formation and resorption.

To verify the effect of glucocorticoid treatment on FUT8 expression and osteogenic response, we treated MC3T3-E1 preosteoblasts with different doses of dexamethasone. We found that mRNA expression of FUT8 was significantly downregulated in response to high-dose dexamethasone stimulation, along with significantly suppressed cell proliferation and osteogenic differentiation, suggesting that high-dose dexamethasone suppresses osteogenesis possibly by downregulating FUT8-mediated core fucosylation of osteogenesis-related glycoproteins. The canonical TGF-β pathway facilitates bone formation (Baffi, Moran & Serra, 2006; Liang et al., 2020; Peters, Wang & Serra, 2017; Wang et al., 2013). Of note, FUT8-mediated core fucosylation is essential for TGF-β binding to TGFBRs (Schachter, 2005; Van Staa, Leufkens & Cooper, 2002), and TGF-β/TGFBRs/Smads signaling is inhibited in glucocorticoid-induced osteoporosis and osteonecrosis (Li et al., 2017; Tao et al., 2017; Xie, Hu & Shi, 2018). Thus, we hypothesized that high-dose dexamethasone suppresses osteogenesis possibly through downregulating FUT8-mediated core fucosylation of TGF-β/TGFBRs and that compensatory expression of FUT8 might reverse the effects of high-dose dexamethasone. Indeed, our results showed that overexpression of FUT8 reversed the inhibitory effects of high-dose dexamethasone on TGF β/Smad signaling, core fucosylation of TGFBRs, as well as TGF-β1 binding to TGFBRs in MC3T3-E1 preosteoblasts. Together, these events facilitate osteogenic differentiation of preosteoblasts, as evidenced by enhanced ALP and Alizarin red staining and mRNA expression of osteogenic markers in MC3T3-E1 in the presence of high-dose dexamethasone.

FUT8-mediated core fucosylation plays an essential role in the activation of TGF-β signaling. Dysregulation of FUT8 blocks the activation of TGF β/Smad signaling (Ng et al., 2018; Wang et al., 2005). Tu et al. have reported that FUT8-mediated TGFBR core fucosylation promotes TGF-β signaling and epithelial–mesenchymal transition, stimulating breast cancer cell invasion and metastasis (Tu et al., 2017). However, whether FUT8 regulates core fucosylation of TGFBRs in preosteoblasts exposed to glucocorticoids remains unexplored. In this study, the results of Co-IP revealed that overexpression of FUT8 dramatically elevated the TGFBR1-LCA/TGFBR1 ratio in the absence of dexamethasone and reversed the effects of high-dose dexamethasone on the ratios of TGFBR1-LCA/TGFBR1 and TGFBR2-LCA/TGFBR2. This finding suggests that FUT8 mediates core fucosylation of TGFBRs in preosteoblasts, thus promoting osteogenesis by activating TGF-β signaling.

Conclusions

In conclusion, high-dose dexamethasone attenuated FUT8 mRNA expression in MC3T3-E1 preosteoblasts. Overexpression of FUT8 reversed the inhibitory effects of high-dose dexamethasone on osteogenic differentiation, TGF-β/Smads signaling, core fucosylation of TGFBRs, as well as TGF-β1 binding to TGFBRs in MC3T3-E1 cells. These findings suggest that overexpression of FUT8 is as a potential therapeutic approach to counteract the side effects of glucocorticoid therapy through activating TGF-β signaling.

Supplemental Information

Supplemental Information 1 Raw data

Click here for additional data file.

We thank Ying Li for her help in the amplification of plasmids.

Additional Information and Declarations

Competing Interests

Author Contributions

Data Availability

The authors declare there are no competing interests.

Zhiming Wu, Tianye Lin and Pan Kang performed the experiments, analyzed the data, prepared figures and/or tables, authored or reviewed drafts of the paper, and approved the final draft.

Zhikun Zhuang, Haibin Wang, Wei He and Qiushi Wei performed the experiments, authored or reviewed drafts of the paper, and approved the final draft.

Ziqi Li conceived and designed the experiments, authored or reviewed drafts of the paper, and approved the final draft.

The following information was supplied regarding data availability:

The raw data are available in the Supplementary File.

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
