# Peer review of "Overexpression of fucosyltransferase 8 reverses the inhibitory effect of high-dose dexamethasone on osteogenic response of MC3T3-E1 preosteoblasts"

_PeerJ, doi:10.7717/peerj.12380_

## Round 0.1 · original submission · Minor Revisions

Please revise your manuscript based on the comments of Reviewers 2 and 3.

When omitting Figure 1, remember to update figure numbers in the text and legends.

In response to the suggestion of evaluating primary osteoblasts, either comply or justify the use of a single cell line (MC3T3-E1).

Reviewer 1 ·

Basic reporting

This manuscript is well written with clear and professional English. The basic knowledge is provided with sufficient details to readers in the introduction to understand the background of this study. The figures are high-resolution and clear. Raw data are provided.

Experimental design

Long-term use of dexamethasone leads to osteoporosis by inhibiting TGF-b. FUT8 is an essential enzyme for the stimulation of TGF-b/smad pathway. This study demonstrated the involvement of TFG-b in osteoporosis induced by high concentration of dexamethasone, potentially by downregulation of FUT8. By overexpressing FUT8 in cells, the authors show that the inhibitory effect of high-dose dexamethasone is reversed, confirmed by the TGF-b binding efficiency and the levels of fucosylation. The hypothesis was well defined. The experiments were well designed with sufficient technical and biological repeats.

Validity of the findings

The data are solid and well interpreted with valid statistics.

Reviewer 2 ·

Basic reporting

no comment

Experimental design

no comment

Validity of the findings

no comment

Additional comments

High-dose and long-term of glucocorticoid treatment may lead to osteoporosis and osteonecrosis. Thus, the identification of key molecules involved in glucocorticoid-induced osteoporosis is important for improving bone regeneration in glucocorticoid therapies. The study showed that overexpression of FUT8 facilitates osteogenesisin dexamethasone therapies by counteracting the inhibitory effect of dexamethasone on TGF-β signaling. The study is interesting, but some concerns need to be addressed.
1. The Figure 1 should be omitted, and the schematic diagram based on research results is supplemented. 2. It is better if the authors check the primary osteoblasts isolated from in vivo experiments.
3. The Figure 2,3, why was the Quantitative real-time PCR measured only at day 3 and not more time points?

·

Basic reporting

In general, the study related to the potential “Overexpression of fucosyltransferase 8 reverses the inhibitory effect of high-dose dexamethasone on osteogenic response of MC3T3-E1 preosteoblasts” is very interesting. The research data submitted is also very sufficient which is accessed by several appropriate approaches. These findings will pave the way for therapeutic strategies to improve bone regeneration in glucocorticoid therapy through activating TGF-β at its receptors.
Some points to note are:
1. What should be criticized is that TGF- β activation at the TGF b superfamily receptor has also been reported to inhibit the differentiation stage from immature osteoblasts to osteoblasts, also inhibits RANKL/OPG which has an impact on decreasing bone mineralization as reported by Wu et al (2016). ) TGF-β and BMP signaling in osteoblast, skeletal development, and bone formation, homeostasis and disease, Bone Research 4(1):16009, DOI 10.1038/boneres.2016.9 (lines525-527). Will the overexpression of fucosyltransferase-8 improve and accelerate bone mineralization? In this research report there is no direct evidence for this purpose.
2. The methods in abstract lines 34-36 require a brief purpose of using the method so as to provide an illustration of the approach used. Not just a listing of the methods during the research. Also at the conclusion of line 50 …osteogenesis in dexamethasone therapies by….. does not match the reported study results.
3. In the introduction, it is written very well and clearly

Experimental design

The methods written in the manuscript has been carried out in detail, clearly and is easy to follow.

Validity of the findings

I have carefully studied the reported data. Data were obtained by standard design with adequate replication. There needs to be a quality improvement on some of the figures.
1. The results of the study, Figure 1 should be discarded because it is only a prediction that will obscure the results of this well-done research. The schematic image is also very unclear regarding the black arrow and the presence of fucosyltransferase 8 either intracellular or extracellular site.
2. Figure 4 panel D should be completed with pEGFP and pEGFP-Fut8 data bands.
3. Figure 5 panel A requires high resolution so that the overlay between DAPI and LCA staining can be seen clearly. Also on panels B and C must be completed with pEGFP and pEGFP-Fut8 data bands.

Additional comments

There are no additional comments

---

## Round 0.2 · accepted · Accept

Thank you for addressing all the issues raised in the review. I am glad to accept your manuscript.

Reviewer 2 ·

Basic reporting

OK

Experimental design

OK

Validity of the findings

OK

Additional comments

NO

·

Basic reporting

After thoroughly reviewing the revised version of the manuscript, I think that there has been a very significant improvement. All answers to questions and clarification of critical points have been well addressed by the author(s). Unnecessary schematic drawings have been removed. Likewise, images that need to be increased in resolution have also been done well. I take up a full explanation related to the completeness of the data to be part of things that need to be clarified. Therefore, I can accept all points that have been finalized and revised in the last version of this manuscript.

Experimental design

The experimental design is apparent and easy to understand

Validity of the findings

The data is very sufficient

Additional comments

No comments